# Intellectual capital and the efficiency of SMEs in the transition economy China; Do financial resources strengthen the routes?

**Guowei Li[1], Zhe Luo[1]\*, Muhammad Anwar[2], Yuqiu Lu[3], Xiantao Wang[3], Xuening Liu[4]**

**1** Civil Aviation Flight University of China, Guanghan, China, **2** College of Economics and Management, Beijing University of Technology, Beijing, China, **3** School of Public Administration, Sichuan University, Chengdu, China, **4** School of Economics, Sichuan University, Chengdu, China

\* sculuozhe@126.com

**Data Availability Statement:** The data are collected through a structured questionnaire from Chinese SMEs where owners and managers of the firms were ensured that the data are used only for

## Abstract

Intellectual capital has been grabbed the attention of researchers due to its momentous role in sustainable competitive advantage and organizational success. There is a growing catalog of related assessments, publications and reviews that display the direct and indirect role of intellectual capital in business success and profitability. Despite the bourgeoning literature, studies have not yet unleashed the influence of each dimension of intellectual capital; human capital, structural capital and customer capital on SMEs' efficiency with financial resources as a moderator. The present study fills the gap and assesses if financial resources strengthen the paths between the dimensions of intellectual capital and SMEs' efficiency. A survey method was used and collected evidence from 264 Chinese SMEs. The findings exhibit that human capital directly enhances SMEs' efficiency but the presence of financial resources as a moderator weakens the influence. However, social capital and customer capital do not directly improve SMEs' efficiency but financial resources reinforce the paths social and customer capital and SMEs efficiency. This research recommends that owners and managers of SMEs need to use their financial resources complementary with structural and customer capital while human capital should be used exclusively.

## Introduction

Steered by globalization and advanced technology, many firms (especially in emerging economies) persistently search to discern drivers that can configure their performance and survival in the dynamic markets. Doubtlessly, a variety of tangible (land, material, capital and technology etc.) and intangible (knowledge, R&D and reputation, etc.) have been recognized that spur business profitability (e.g, [1–3]). Considering the nature and characteristics of Small and Medium Enterprises (SMEs), as they have a deficiency of resources, lack of support and smallness (e.g. [4], typically make them unable to invest a huge amount of money in risky and physical assets. This sensation encourages them to recognize alternative opportunities (less risky, convenient and easy to adopt) in order to enhance their survival in the markets [5]. Since the last decade, studies have shown their interest in discovering factors that can significantly boost

the research purpose and will not be available to all the public. Restriction from the ethics committee of Sichuan University is applied for data access. Hence, the data (related to tables and figures) can be obtained on request from the ethics committee of Sichuan University: hassan2018@stu.scu.edu. cn.

**Funding:** This study received funding for the development of rural human capital in Sichuan province under the background of rural revitalization project of Sichuan science and technology plan (soft science) (serial number:18rkx1285) and education results.

**Competing interests:** NO authors have competing interests.

and the path to SMEs performance such government support [6], knowledge sharing [7], creativity [8], innovation [9]and IT capabilities [10] and entrepreneurial orientation [11] etc. Studies have also discussed the role of intellectual capital (IC) (human capital, customer capital and structural capital) in firm competitive advantage and performance in emerging and developed economies [12, 13] Also, studies have assessed the direct and indirect influence of IC on performance [14–16]. Nevertheless, it is still unclear "how each dimension of IC; human capital, customer capital and structural capital contributes to SMEs efficiency"? This research is an attempt to examine the influence of each dimension of IC on SMEs' efficiency. Though, IC may not significantly contribute to efficiency as some studies pointed out the indirect influence of IC on performance [17–19]. In this regard, SMEs may need some other factors that can support and transpire IC into high efficiency. For instance, Cheng and Krumwiede [20] argue that all the intangible factors do not significantly enhance performance in environmental uncertainty but firm resources required to configure the situation. We argue that financial resources can strengthen the link between IC and SMEs' efficiency. This research, on the other hand, examines the moderating role of financial resources between each dimension of IC and SMEs' efficiency.

There are ample evidence that confirm the significant role of IC in the financial performance of enterprises. For instance, [21] argued that efficient use of IC facilitates SMEs in exploiting growth opportunities and gaining high financial performance. [22] demonstrated that the dimensions of IC; human capital, structural capital and relationship capital are very vital for efficiency in the banking industry in the long run. Alhassan and Asare [23] also revealed that human capital and capital employed to boost the productivity of banks in emerging economies. Similarly, Oppong and Pattanayak [24] revealed that productivity and competitive advantage of an enterprise are no longer dependent on tangible resources but heavily rely on the intangible resource, particularly IC. This notion has been supported by several authors. For instance, Kamath [25] scrutinized that human capital is the most significant predictor of productivity in Indian business organizations. Additionally, Mondal and Ghosh [26] state that the banking industry enhances its efficiency through human capital. However, some studies shed light on the importance of structural capital in the business industry. Bontis, Janošević [27] scrutinized that many business firms rely on structural capital because it articulately works for them in improving productivity. Soriya and Narwal [28] indicated that many organizations configure their employee productivity through structural capital.

There are several reasons why financial resources can significantly moderate the relationship between IC and efficiency. For instance, considering the high failure ratio of SMEs across the globe, studies have reported the major reason of lack of finance (e.g [29, 30]). Having enough finance enables SMEs to boost their operational activities effectively. For instance, Fonseka et al [31] demonstrate that out of many resources, financial capital is the most prominent for SMEs in emerging economies. Moreover, Cooper, Gimeno-Gascon and Woo [32] also describe that human and financial capital help newly established ventures as a shield to respond to the external threats and changes. Thus, financial resources enable firms to get benefits from the IC and transform their skills into the best use. Additionally, some firms are unable to enter into profitable markets due to a lack of finance. Thus sufficient finance encourages them to seize the benefits of new markets [31]. Adequate financial capital facilitates firms in opportunity recognition that is helpful for efficiency e.g [33] Some firms are unable to effectively use their skills and resources [34]. In this case, financial capital, as a driver can assist managers to reorganize the resources and skills in a way to gain high profit [35].

This research aims to explore the role of financial resources in firm efficiency as well as how financial resources strengthen the relationship between IC and SMEs' efficiency. Alternatively, as described earlier, this research aims to recognize less risky sources and resources that enable

SMEs to gain efficiency in the turbulent markets. This research assesses the Resource Base View (RBV) theory which tells the role of tangible and intangible resources in firm competitive advantage and superior performance [36]. More interestingly, RBV theory gives more emphasis to intangible resources over tangible in terms of high performance [2, 36]. In this perspective, this research argues that intangible resources e.g. IC can give a greater advantage over tangible resources especially in emerging economies. This research is based on SMEs operating in emerging market China where more than 95% of firms are SMEs. The country heavily relies on SMEs but still, they face big pressure and a high failure ratio (e.g, [37]). The findings of this research facilitate owners, managers and policy-makers to give due attention to the survival and performance of SMEs, so they will enable them to make prominent contributions to economic growth, GDP and propensity. There is a great need for financial institutions and SMEs banks in the country in order to provide interest-free loans or where enterprises can borrow a loan at a lower interest rate.

## Theoretical background and hypotheses development

Efficiency is defined as "using minimum input to gain maximum output"

There are three major types of efficiency; technical efficiency, economies of scale, and technical change [38].

a. Technical Efficiency: Farrell [39] operationalized the concept of technical efficiency as well as discussed what factors lead to inefficiency in production. Technical efficiency refers to the ratio between actual output and the maximum output of the enterprises that could produce with the set of inputs and technology. In general, this type of efficiency based on the resources possessed by a firm. Hence, large firms are more efficient from a technical perspective because they enjoy sufficient resources [40] while small firms have been criticized for being a deficiency of resources that hamper them from attaining technical efficiency [41]. SMEs across the globe face a shortage of resources [2]. Therefore, they need to strengthen their IC strategy in order to compete in the market and gain technical efficiency.

b. Economies of Scale: It is sometimes called cost advantage where a firm increases its production that minimizes the cost. When an enterprise produces more—it becomes more efficient and thereby lowering the costs. This type of efficiency is very popular in both small and large firms as both firms try to gain maximum benefits at a lower cost. However, in particular, small firms are more cost concentrated because of the deficiency of resources [2]. They intend to reduce different types of costs, resulting in high profitability [42].

c. Technical Change: This type of productivity explains production differential. It measures the change in efficiency between the current and next period (t–t+1). It is not necessarily technological change but can be occurred by the change in regulation, quantities of inputs and price of inputs etc. This type of efficiency is also very crucial in SMEs because of the internal structure that is not restricted they often modify their strategic posture when they see any benefits or loss [43]. They are more inclined towards change and innovation, thereby emphasizing on high profitability and outputs [9].

### Intellectual capital SMEs efficiency

RBV theory [36] sheds light on the prominence of tangible and intangible resources in the realism of sustainable performance in a turbulent market. Indeed both the resources are very crucial for high performance, productivity and competitive advantage [44]. However, recent studies have claimed that tangible resources are no longer work for productivity and

sustainable competitive advantage in SMEs but intangible factors are more crucial (e.g., [24]; [37]. This notion is supported by several studies [45]; [2]. IC is deemed an intangible factor that has been remained a prominence piston in SMEs for profitability, sustainable positon and exploiting opportunities [21, 46]). The reasons behind the significance of intangible resources (hereby IC) in SMEs are deficiency of resources in SMEs, smallness and lack of capacity to invest in tangible resources [45]. For instance, Kraja [47] conducted a study check the comparative importance of tangible and intangible resources and revealed that intangible resources are more powerful for SMEs' performance. Khan, Yang [2] also argued that investment in intangible resources enables emerging SMEs to gain sustainable competitive advantage and superior performance in the dynamic markets. Indeed, intangible resources are less expensive but give more advantages [37]. Therefore, SMEs emphasize intangible resources, especially IC to enhance their position in the market [48]. Human capital—a dimension of IC facilitates enterprises in efficient utilization of resources, thereby resulting in maximum benefits [49]. Investment in human capital is one of the most critical decisions of enterprises because it significantly leads to productivity [50]. Kong and Kong [51] also dissected that human capital significantly and positively increases the productivity of private and public enterprises in China. Haris, Yao [52] displayed that structural capital significantly underpins the profitability of the banking sector in the emerging market Pakistan. According to [53] (p. 139–140) "Structural capital is, what Romer would call "ideas", because it can be replicated at a large scale and low costs. This implies that structural capital is the true force behind productivity. Furthermore, Leal-Millán, Roldán [54] describe that customer capital is the key to sustainability and performance in a turbulent environment because customer knowledge gives superior advantages. To summarize, all the considerable ample evidence indicates that IC—being an intangible asset plays a crucial role in the productivity of SMEs that is the main theme of the RBV theory.

Even though IC has been growing since the last several decades but still universal definition is not yet confirmed, rather it involves more rigorous conceptualization in theory and practice. Most managers and scholars have vague concepts about how to manage invisible resources; human capital, customer capital and structural capital [55, 56]. According to Stewart [57], IC as the intellectual material of information, knowledge, experience and intellectual property that can be utilized to create wealth. Studies have discussed various dimensions that can encompass IC such as human capital, relational capital, customer capital, structural capital, organizational capital and social capital etc. (e.g [58, 59]). However, many studies agreed on the three dimensions; human capital, customer capital and relational capital that are used in this research.

IC is an essential part of firm activities related to financial and non-financial. It affects the internal processes and reporting system of firms that are aligned to profitability and effectiveness [60]. According to a 'resource-based view' of competition, IC is considered as an important source of competitive advantage. In prior studies, efficiency is measured in two dimensions; technical efficiency and cost-efficiency. IC as knowledge, intellectual property, information, analytical skills, competencies and expertise that are used by a company to enhance its competitive advantage that ultimately influences shareholders' wealth [61]. In the current turbulent markets, companies face a dramatic change and high pressure from the external environment. In these circumstances, IC such as human capital, structural capital and relationship capital is the best strategy sustain for the long run and gain high efficiency [62]. IC is a good type of activity that helps the organization to use the knowledge effectively to gain high financial performance and efficiency [63]. Though some studies have scrutinized that all the dimensions of IC are not significantly related to firm competitiveness and efficiency. For instance, Yaseen, Dajani and Hasan [64] claimed that only relational and structural capital significantly enhance a firm competitive position. Besides,

Knowledge in the economy or an organization can be circulated through human, structural or relationship capital—components of IC. Well managed IC can help firms to gain success and competitive advantage in the markets that cannot be achieved by competitors and industry rivals [65]. Firms often use HRM practices for smooth operation. However, IC can be a prominent factor for highly innovative and competitiveness [66]. IC is very necessary for manufacturing firms as they are engaged in manufacturing activities that required high analytical and intellectual abilities. IC help manufacturing firms to redesign their internal structure and strategic process which in turn facilitate firms to gain efficiency [20]. A firm efficiency can be improved by several factors but intangible resources (especially IC) can significantly enhance efficiency as compared to other factors [67]. Firms, especially in emerging markets, face several problems including lack of resources and capabilities. Hence, they adjust the shortcomings through IC to gain a highly competitive position and effectiveness over a longer period [68]. In a competitive market, where a firm is trying to be highly innovative and successful, it needs efficiency that can be gained through social capital [69]. It is argued that firms should emphasize on intangible resources (e.g., IC) to achieve effectiveness and innovative performance in a dynamic market [20]. All the dimensions of IC (human, structural and relational capital) are significantly positively related to firm efficiency [70]. Recently, IC has deemed an essential intangible asset for business organizations, especially in those business industries that are characterized by advanced technology and high intensive knowledge capital. In addition, IC is used to configure the business's success and to encourage innovativeness, creativity, value creation and competitive edge among firms [71]. To summarize, IC provides capabilities and resources to build a competitive advantage for organizations. A firm without using IC may not be able to gain sustainable competitive advantage and superior performance in a specific market or industry and without a competitive advantage, a firm can quickly exit the industry [48]. Therefore;

*H1. Human capital significantly spurs SMEs efficiency*

*H2. Structural capital significantly spurs SMEs efficiency*

*H3. Customer capital significantly spurs SMEs efficiency*

## The moderating role of financial resources

Organizations possess various resources e.g. tangible and intangible that are either directly or indirectly influenced their performance, competitive advantage and survival [72]. IC is regarded as an intangible asset within organizations that can spur their performance [73, 74]. It is argued that human capital significantly contributes to firm value and success but the relationship can be affected by a firm internal and external factor [75]. Considering the indirect influence of IC toward performance and profitability, we argue that financial resources can significantly tight the link between IC and firm efficiency. Financial capability is considered the ability or capacity of a business to use financial resources as a medium of exchange for other productive resources [76]. The RBV theory suggests that a venture needs tangible and intangible resources to acquire sustainable status in the market [36]. Hence, we perceive that IC and financial resources should work combine for enterprises to seize a competitive position and satisfactory efficiency. For instance, Tseng, Lin and Yen [77] claimed that venture processing activities are influenced by both IC and financial capital. Similarly, Yan and Ning [78] also scrutinized that the relative importance of IC and financial capital in business values and argued that a firm should not exclude any of these because both are very crucial for high corporate value. It is doubtless that intangible resources are essential for SMEs' growth and

survival in transition and emerging economies. However, mere intangible resources do not boost operational activities of SMEs, they must have satisfactory finance to recognize new opportunities to enhance their innovative and financial performance [79]. The efficient use of financial resources saves ventures from wastage of capital in different investment activities and enables firms to utilize the resources properly [80]. Other resources and capabilities facilitate business activities effectively. However, financial resources and quality financial management boost operational activities in the business sector [81]. When a venture tries to recognize new opportunities in a competitive market, it needs financial and non-financial resources. Though the resources such as market and entrepreneurial orientation facilitate the discovery of opportunities for earning profitably, but financial resources amplify the paths towards high earning [82]. For instance, consider the theory of arbitrage finance [83], that criticizes the role of external capital due to a low level of performance and gives worth to internal financial resources due to speed progress during high environmental uncertainty. Hence, we align our model to the theory because our model also sheds light on internal capital that can configure other resources (e.g. IC) for business operations. This argument is supported by Kaleka [84] who scrutinized that a firm with greater financial resources enjoys superior performance due to the unique role of the capital in effectively constituting other resources.

For instance, Penrose, [85] securitizes that a firm's resources such as financial capital, talented managers and knowledge etc. are the key inputs into the production process, value creation and high efficiency. Indeed enough resources are very crucial for SMEs' success. However, financial capital is deemed very important in spurring a firm's efficiency and growth [86]. A firm with greater access to financial resources enables them to use their internal resources smoothly and articulately that can give high competitiveness and profit [87]. Financial resources not only enable firms to recognize and exploit the internal opportunities but also help them to reconfigure their internal process to react external opportunities appear in international markets to improve their profit [88]. Adequate finance enables firms in access to unique and scarce resources that are essential for growth and smooth running of operational activities [89].

Having strong financial resources can encourage firms to get into a more risky situation which in turn provide higher financial benefits [90]. IC is created through the exchange and combination of intangible resources that may be characterized as explicit or implicit knowledge within organizations. Consequently, organizations need to differentiate themselves and perform tasks and actions differently in order to succeed in the markets [91]. Hence, a sustainable competitive advantage does not come automatically by offering final products and services to the customers but it comes from the resources that produce them. Resources here also deemed financial and other that can reconfigure IC towards high performance and competitiveness. As pointed out by Hunt and [92], competitive advantage will not continue unless a firm uses its resources efficiently and effectively to deliver values to a specific segment in the markets. It posits that a firm needs to develop value-creating strategies from its sources for improving sustainable efficiency [36, 93]. One significant factor, especially in SMEs, is financial capital that does not only help to a successful venture in the long run but also shields them from unexpected loss [32]. Therefore, we argue that an enterprise needs adequate finance that should be used combined with IC to gain desirable efficiency in the competitive market. Consequently,

*H4. Financial resources strengthen the association between human capital and SMEs' efficiency in the way that the association will be stronger when enterprises have sufficient financial resources.*

*H4. Financial resources strengthen the association between structural capital and SMEs' efficiency in the way that the association will be stronger when enterprises have sufficient financial resources.*

*H4. Financial resources strengthen the association between customer capital and SMEs' efficiency in the way that the association will be stronger when enterprises have sufficient financial resources.*

The conceptualized model is illustrated in Fig 1.

## Methodology

### Sample and data

This research is based on SMEs operating in the Transition economy China. We targeted SMEs from three major cities; Beijing, Shanghai and Shenzhen because majority ventures have their head offices in these regions. Unlike the email survey which gives a lower response rate, we used a hard copy approach for data collection. It is difficult to estimate the number of enterprises in these regions. Hence we used the nonprobability method and followed a convenience sampling method for the collection of the data. A total of 600 questionnaires were distributed among the firms (200 in each city). We requested owners and top managers as they are the people who know the financial position and strategic activities of their firms. Additionally, where managers were not aware of exact financial figures, they recommended us to ask financial managers regarding efficiency and financial outcomes. We asked them that the survey is volunteer to be filled. To reduce social desirability bias, we have clearly mentioned in the covered letter of the survey that the data of this survey is exclusively used for research analysis, and the information will not be shared elsewhere. Since Chinese managers face difficulty in understanding an English version questionnaire, therefore we translated the questionnaire in the Chinese language that was approved by the committee for research. Once the questionnaire was approved by the committee of Sichuan University China, we started data collection from the enterprises. In the questionnaire, it was ensured that the data (collected through this questionnaire) will be used only for research purpose and the firm information will not appear

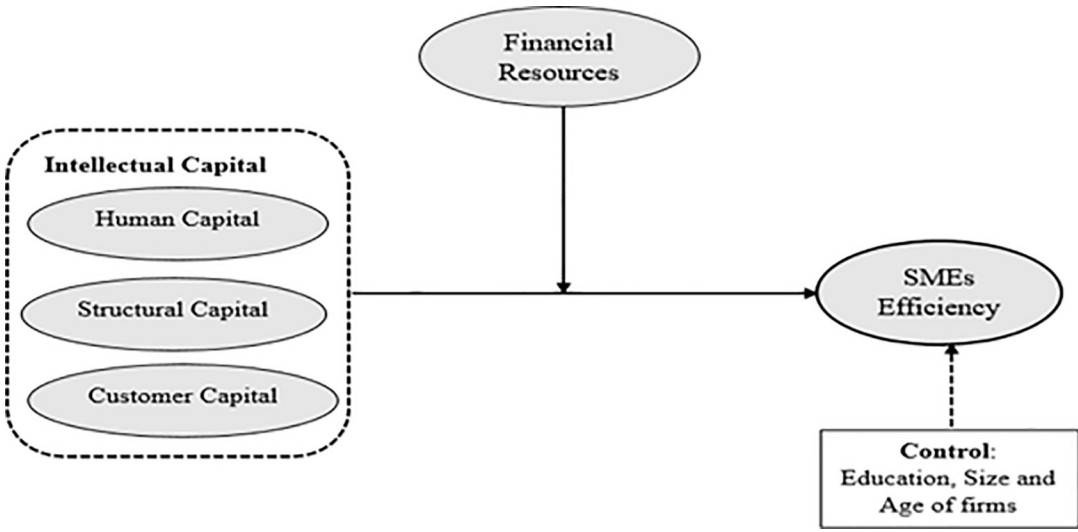

**Fig 1. Model of the research.**

anywhere else. We followed a hard copy approach as it gives a desirable response rate as compared to an online survey. Managers and owners were asked to fill the survey voluntarily and during the busy schedule, we requested to fill the survey later. We received back 285 questionnaires but some of these questionnaires were incorrectly filled. So we dropped 21 questionnaires and included only 264 usable questionnaires in the analysis of this research. The response rate was 44%. The demographics detail of the firms who participated in this research are presented in Table 1. Out of total of 264 owners/managers, 43 respondents have intermediate and a low level of education, 78 owners/managers who have completed a Bachelor's degree, 96 were master qualified while only one respondent was Ph.D. qualified. We surveyed manufacturing, trading and services ventures for this research. In our sample, the majority of ventures (139) were engaged in manufacturing activities, 61 enterprises were from the trading sector and 64 were from services businesses. Our sample shows that 93 ventures have started their operation since the last 10 years and less, 89 enterprises have operated since 11 to 20 years and 82 firms were working for more than 21 years.

## Measurement of variables

**Intellectual capital.** IC in this study is used as an independent variable. To derive useful insights, we used each dimension of IC; human capital, customer capital and structural capital. Prior studies have used various dimensions to measure IC. The items for human capital, customer capital and structural capital were adopted from Liu [14] and Jain, Vyas and Roy [48] respectively.

**Financial resources.** financial resources indicate a firm capacity to have finance and capital to be used for the smooth running of operational activities. To measure FA, we used five items adopted from the prior study Memon, An and Memon [79].

**Table 1. Background of the SMEs.**

| Particular | Frequency | Percent |
|---|---|---|
| Qualification | | |
| 1. intermediate and less | 43 | 16.3 |
| 2. Bachelor | 78 | 29.5 |
| 3. Master | 96 | 36.4 |
| 4. MS / MPhil | 46 | 17.4 |
| 5. PhD | 1 | 0.4 |
| Industry | | |
| 1. manufacturing | 139 | 52.7 |
| 2. trading | 61 | 23.1 |
| 3. services | 64 | 24.2 |
| Size of the SMEs | | |
| 1. 20–100 employees | 57 | 21.6 |
| 2. 101–200 employees | 51 | 19.3 |
| 3. 201–300 employees | 50 | 18.9 |
| 4. 301–400 employees | 61 | 23.1 |
| 5. 401 to 500 employees | 45 | 17.0 |
| Age of the SMEs | | |
| 1. 10 years and less | 93 | 35.2 |
| 2. 11–20 years | 89 | 33.7 |
| 3. 21 and above years | 82 | 31.1 |
| Total | 264 | 100 |

**Firm efficiency.**   in large firms, researchers typically use a formula output/input to measure efficiency.

SMEs do not publish their financial information with the general public which creates the problem in measuring their performance and efficiency. In this regard, researchers have recommended self-reported measures because they have several advantages over objectives measures;

- It is difficult to obtain a financial statement and financial records of SMEs because they do not publicly share their financial information [94].

- Several studies have confirmed a significant match in the results of self-reported and archived data used for performance [95, 96, 97].

- Semrau, Ambos [98] claimed that self-reported measures give better results as compared to archived data in emerging economies. Therefore, we focused on a self-reported measure for the efficiency of the SMEs.

To measure efficiency, owners/managers were asked how they use the least resources for maximum outputs in terms of return on investment, return on assets and return on sales, technology on production etc.

The items were measured using five points Likert scales illustrating strongly disagree 1 to strongly agree 5. However, for measuring efficiency, we used strongly declined 1 to strongly improved.

## Control variables

Control variables are used to attenuate the chances of spurious results in a research study. In SMEs research, the nature of the industry, age and size of the enterprises and educational background of top managers and owners are suggested by several studies to be controlled [95]; Khan et al., 2019; [45]. Following the suggestions of these studies, we also control these particular variables in our structural model when testing the hypotheses. The nature of the industry is a categorical variable, hence we created group difference analysis to check it as a controlling factor. We created three groups; manufacturing1, trading2 and services3 and then compared the results of each group with others. We did not realize a significant difference between the groups, hereby dropped the nature of industry from the control list because of a minor role. However, the age and size of the enterprises and educational background of the owners/managers are the continuous variables that are used as a control in the structural model. Our results displayed that the size of the enterprises does not play a significant role while the age of the firms and qualification of the top managers and owners indicate a significant role in the hypothesized model.

## Data analysis and results

We used SPSS to check the Mean ($M$), Standard Deviation (S.D.), multicollinearity and normality of the data. Our study shows (see Table 2) that the $M$ and S.D. values of human capital are 3.62 and 0.48 respectively. Similarly, $M$ and S.D. values of structural capital, customer capital, financial resources and SMEs efficiency 3.49(0.51), 3.68(0.45), 3.21(0.58) and 3.21(0.51) respectively. Considering the skewness and kurtosis, our data are normally distributed because all the constructs have acceptable skewness and kurtosis values ±2 as suggested by George [99]. Moreover, there is no threat of multicollinearity in our study because there is no exceed value (above 3) of Variance Inflation Factor (VIF) and lower value (below 0.10) of tolerance in our sample [100]. Hence, the descriptive statistics of the data are satisfactory and acceptable.

**Table 2. Descriptive statistics.**

| Variables | Mean | Std. Deviation | Skewness | Kurtosis | VIF | Tolerance |
|---|---|---|---|---|---|---|
| HC | 3.6220 | 0.47920 | -1.409 | 1.894 | 1.184 | 0.845 |
| SC | 3.4924 | 0.50796 | -0.600 | 0.090 | 1.181 | 0.846 |
| CC | 3.6768 | 0.44974 | -0.768 | 1.779 | 1.373 | 0.728 |
| FR | 3.2109 | 0.58149 | 0.469 | 0.388 | 1.086 | 0.921 |
| FE | 3.2107 | 0.51423 | -0.181 | -1.044 | - | - |

FE = Firm Efficiency, HC = Human Capital, SC = Structural Capital, CC = Customer Capital, FR = Financial Resources

## Correlation

The relationship between the factors is given in Table 3. Scholars do not accept or reject hypotheses on the results of correlation values but it just initially supports the findings. It illustrates that HC, customer capital and financial resources are significantly positively related to efficiency (r = 0.175, r = 0.187 and r = 0.175) respectively. However, the association between structural capital and efficiency is not significant (r = 0.091). The correlation results confirm that the sample has no problem with multicollinearity because all the values are below 0.80 [101].

## Common method variance

Data gathered through closed-ended questions, from the same respondents and at the same time face common method variance problem [102]. Hence, it is essential to test the common method variance problem in the data set. We used Harman's single factor test in SPSS to check the threat of common method variance. Our findings revealed only five factors of which the first factor displayed only 24.15%. This variance is in the acceptable range (below 50%) as recommended by [102]. However, Harman's single factor test is criticized for lack of validity. Therefore, we tested the impact of a common latent factor on the measurement model and compared the results of the models (one with a common latent factor and one without the common latent factor). We discovered that the fitness of the old model (without the common latent factor) are significantly better than the new model (with the common latent factor) which clarifies that our sample is free of common method variance.

**Table 3. Correlations.**

| Variables | education | Size | Age | HC | SC | CC | FR | FE |
|---|---|---|---|---|---|---|---|---|
| education | 1 | | | | | | | |
| Size | -0.014 | 1 | | | | | | |
| Age | 0.121 | 0.084 | 1 | | | | | |
| HC | 0.027 | -0.072 | -0.062 | 1 | | | | |
| SC | 0.074 | 0.041 | 0.027 | 0.202** | 1 | | | |
| CC | 0.099 | -0.132* | 0.050 | 0.389** | 0.376** | 1 | | |
| FR | 0.122* | -0.013 | 0.148* | 0.124* | 0.189** | 0.264** | 1 | |
| FE | 0.181** | 0.074 | 0.237** | 0.175** | 0.091 | 0.187** | 0.175** | 1 |

*. Correlation is significant at the 0.05 level (2-tailed).

**. Correlation is significant at the 0.01 level (2-tailed). FE = Firm Efficiency, HC = Human Capital, SC = Structural Capital, CC = Customer Capital, FR = Financial Resources

## Non-response bias

Early response and late response in a data set can cause non-response bias which can affect results [103]. There were 157 early responses and 107 late responses (after they had given a reminder) in our sample. After calculating mean values of the variables; human capital, structural capital, customer capital, financial resources and the efficiency of SMEs, we applied the T-test to assess if there is any non-response bias in the data. We compared the results of the two groups (late and early responses), we realized that the P-value of T-tests (greater than 0.05) did not indicate a significant difference between the groups, hereby confirmed the absence of non-response biases.

## Validity and reliability

The validity (convergent and discriminant) and reliability (see Fig 2) of the constructs are checked through confirmatory factor analysis in AMOS. However, first, we guaranteed the fitness of the model and revealed that CMIN/Df = 2.804 provided satisfactory value (e.g. below 3) as per the suggestion of Hinkin [104]. GFI = 0.81, CFI = 0.87, TLI = 0.85 and NFI = 0.81 displayed acceptable values as followed by previous studies [104, 105]. Additionally, RMR = 0.023 and RMSEA = 0.079 are desirable (below 0.09) as argued by Steiger [105]. We found (see Table 4) that all the standardized factor loadings of the items (after divided by the average number of items toward each factor) provided satisfactory values for convergent validity (above 0.50) and discriminant validity (above 0.70, after taking the square root of AVE). Finally, Composite Reliability (C.R.) was assessed which exhibited pleasing value (above 0.70) as advocated by Bagozzi [106].

## Structural model

A structural model was applied in AMOS for testing the hypothesized model that is given in Fig 3. The main reason to apply a structural model is that it allows researchers to test all the hypotheses in a single [107]. Hence, we tested all the paths (direct as well as interacted) via a single structural model.

We found (see Table 5) that human capital significantly contributes to SMEs efficiency ($\beta$ = 0.163, C.R = 2.705, P = 0.007) that supported H1. Structural capital does not significantly affect SMEs efficiency ($\beta$ = -0.089, C.R = -1.560, P = 0.119) which did not support H2. Customer capital also did not display a significant influence on SMEs efficiency ($\beta$ = 0.049, C.R = 0.766, P = 0.444) that did not favor H3. Considering the moderating role of FR, our findings exhibited that financial resources as a moderator significantly reduce the influence of human capital on SMEs efficiency ($\beta$ = -0.030, C.R = -2.879, P = 0.004) which opposed the proposed hypothesis 4. However, our study confirmed that financial resources significantly strengthen the path between structural capital and SMEs efficiency ($\beta$ = 0.037, C.R = 3.242, P = 0.001) as well as between customer capital and SMEs efficiency ($\beta$ = 0.033, C.R = 3.131, P = 0.002) that supported H5 and H6 respectively. All the control factors; educational background, size and age of the enterprises exposed a significant role in the model. R square revealed that only 19% variance in the SMEs' efficiency is explained by IC in the presence of the moderating role of financial resources as well as the control factors.

## Interaction terms

Since we have a moderator (FR) between IC and SMEs efficiency in the model. We used unstandardized beta, *M* and S.D. of the factors to draw the interaction term. Fig 4 shows that low and high financial resources do not strengthen the path between human capital and EF.

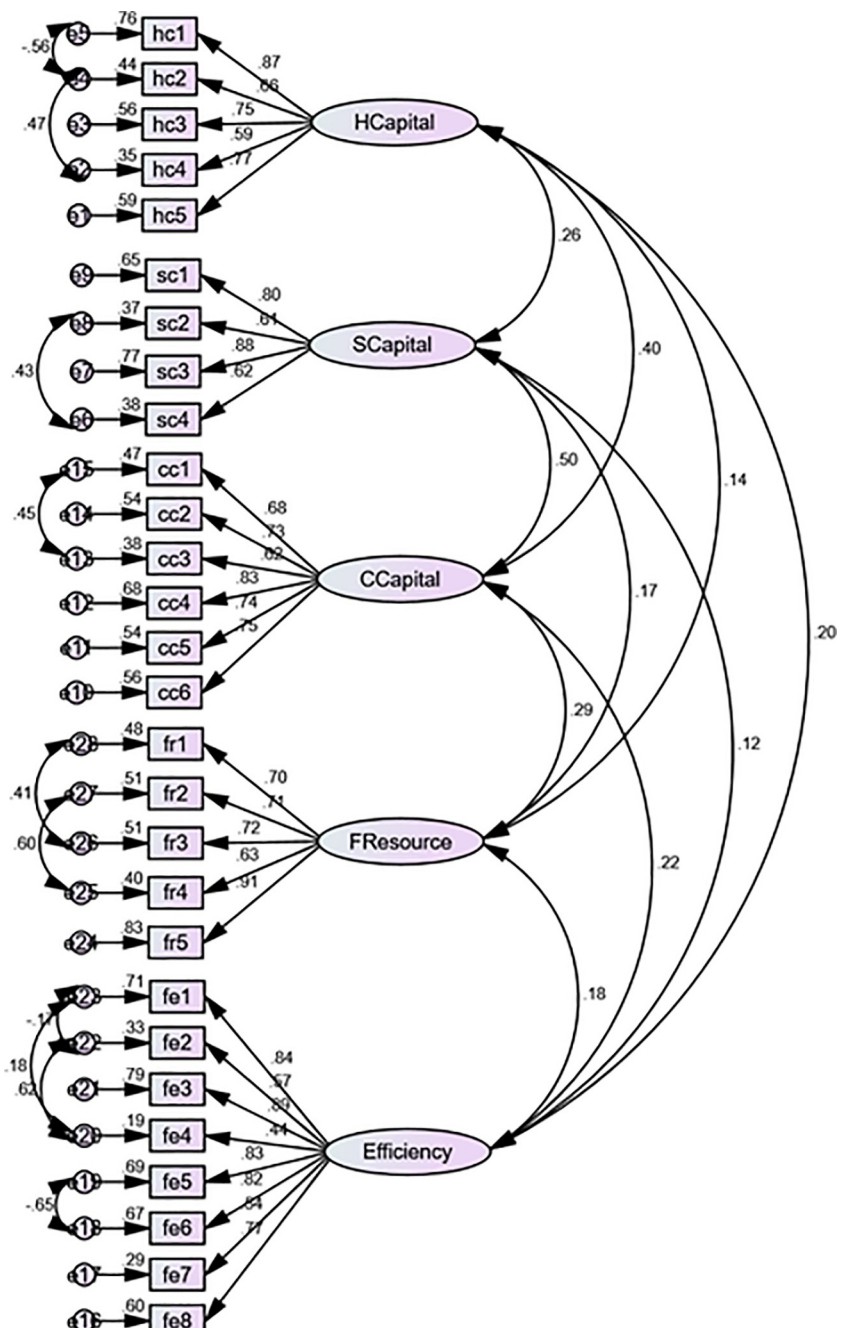

**Fig 2. Measurement model.**

However, high financial resources reduce the influence of human capital on efficiency even if there is high human capital with a venture. The P-value for the interaction term of HCxFR is significant 0.004 (see Table 5, opposite to HCxFR) which confirms that financial resources significantly moderate the link between human capital and efficiency of SMEs.

Fig 5 shows that low financial resources slightly strengthen the path while high financial resources significantly strengthen the link between structural capital and SMEs' efficiency. The P-value of the interaction term 0.001 displays the significant role of financial resources between structural capital and the efficiency of SMEs.

**Table 4. Validity and reliability.**

| Variables | Estimate | AVE | $\sqrt{AVE}$ | C.R |
|---|---|---|---|---|
| Human Capital | | 0.54 | 0.74 | 0.85 |
| hc5 | 0.77 | | | |
| hc4 | 0.59 | | | |
| hc3 | 0.75 | | | |
| hc2 | 0.66 | | | |
| hc1 | 0.87 | | | |
| Structural Capital | | 0.54 | 0.74 | 0.82 |
| sc4 | 0.62 | | | |
| sc3 | 0.88 | | | |
| sc2 | 0.61 | | | |
| sc1 | 0.80 | | | |
| Customer Capital | | 0.53 | 0.73 | 0.87 |
| cc6 | 0.75 | | | |
| cc5 | 0.74 | | | |
| cc4 | 0.83 | | | |
| cc3 | 0.62 | | | |
| cc2 | 0.73 | | | |
| cc1 | 0.68 | | | |
| Financial Resources | | 0.55 | 0.74 | 0.86 |
| fr5 | 0.91 | | | |
| fr4 | 0.63 | | | |
| fr3 | 0.72 | | | |
| fr2 | 0.71 | | | |
| fr1 | 0.70 | | | |
| SMEs Efficiency | | 0.53 | 0.73 | 0.90 |
| fe8 | 0.77 | | | |
| fe7 | 0.54 | | | |
| fe6 | 0.82 | | | |
| fe5 | 0.83 | | | |
| fe4 | 0.44 | | | |
| fe3 | 0.89 | | | |
| fe2 | 0.57 | | | |
| fe1 | 0.84 | | | |

AVE = Average Variance Extracted, it display convergent validity, CR = Composite Reliability,
$\sqrt{AVE}$ = Discriminant Validity

Fig 6 displays that both low and high financial resources significantly improves the influence of customer capital on SMEs' efficiency. However, high financial resources show more strength in the influence of customer capital on SMEs' efficiency. The P-value of the interaction term CCxFR is 0.002, hereby confirms that financial resources significantly strengthens the association between customer capital and SMEs' efficiency.

## Robustness checks

We also performed a robust check-in SPSS by applying regression analysis. Our findings displayed some differences in terms of the interaction results as compared with the structural

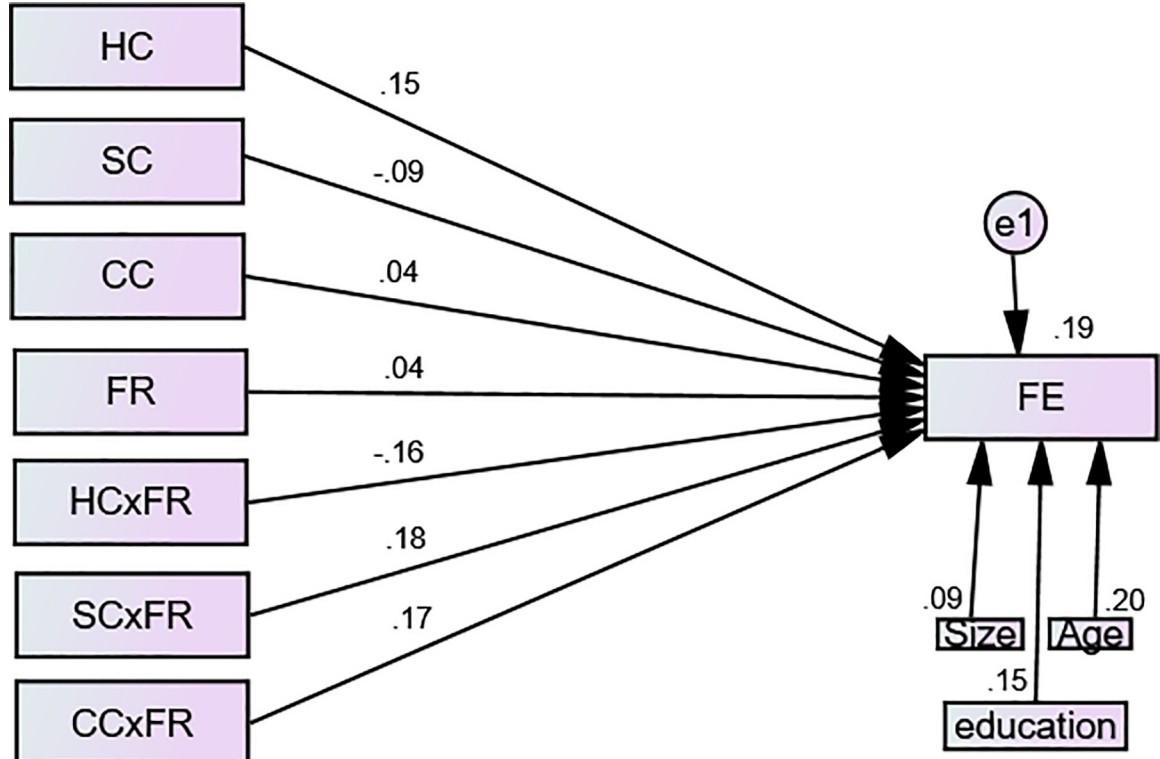

**Fig 3. Structural model.** FE = Firm Efficiency, HC = Human Capital, SC = Structural Capital, CC = Customer Capital, FR = Financial Resources.

model because, in the regression model, two insignificant interactions; HCxFR and CCxFR were reported as shown in Table 6.

## Online interviews

We conducted an in-depth interview with five managers/owners of the SMEs to enhance validity of the results and especially to know if there is social desirability bias in the data. We asked the following questions in the interview to get open ended answers.

**Table 5. Hypotheses testing.**

| Hypotheses | | | Estimate | S.E. | C.R. | P |
|---|---|---|---|---|---|---|
| H1. FE | <— | HC | 0.163 | 0.060 | 2.705 | 0.007 |
| H2. FE | <— | SC | -0.089 | 0.057 | -1.560 | 0.119 |
| H3. FE | <— | CC | 0.049 | 0.064 | 0.766 | 0.444 |
| H4. FE | <— | HCxFR | -0.030 | 0.010 | -2.879 | 0.004 |
| H5. FE | <— | SCxFR | 0.037 | 0.011 | 3.242 | 0.001 |
| H6. FE | <— | CCxFR | 0.033 | 0.011 | 3.131 | 0.002 |
| FE | <— | FR | 0.037 | 0.050 | 0.753 | 0.452 |
| FE | <— | Education | 0.082 | 0.030 | 2.735 | 0.006 |
| FE | <— | Size | 0.033 | 0.021 | 1.582 | 0.114 |
| FE | <— | Age | 0.127 | 0.035 | 3.610 | 0.001 |

FE = Firm Efficiency, HC = Human Capital, SC = Structural Capital, CC = Customer Capital, FR = Financial Resources

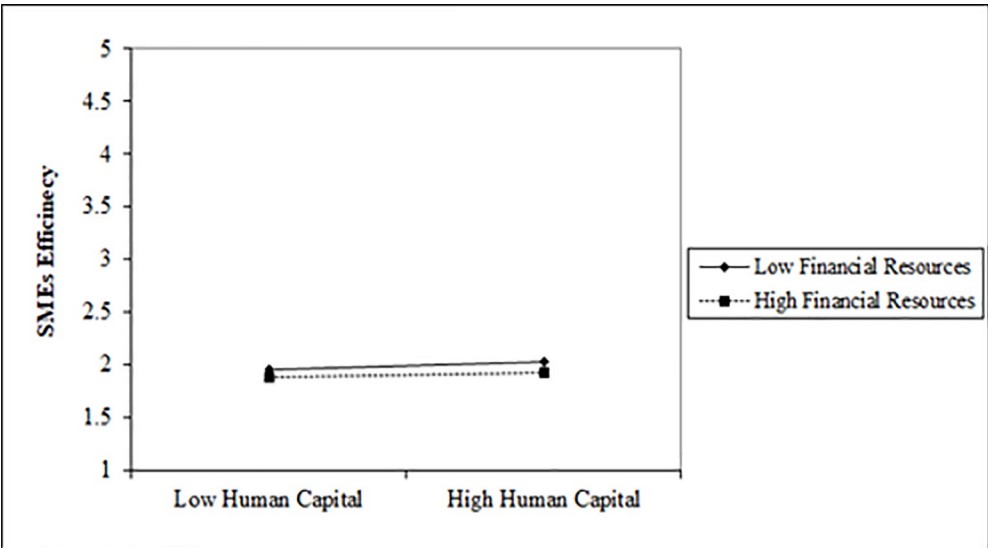

**Fig 4.**

1. Elaborate your IC strength in terms of human capital, structural capital and customers' capital and how it benefits your firms, especially performance and efficiency?

2. Ans: Three managers shed light on the importance of their IC for high performance and efficiency while two firms replied that their IC averagely contribute to their performance.

3. How your existing financial resources moderate the path between IC and efficiency?

4. Ans: Four managers said that they get high benefits of IC when financial resources work as a moderator while one manager said that my firm faces financial constraint, and we are unable to get desirable efficiency.

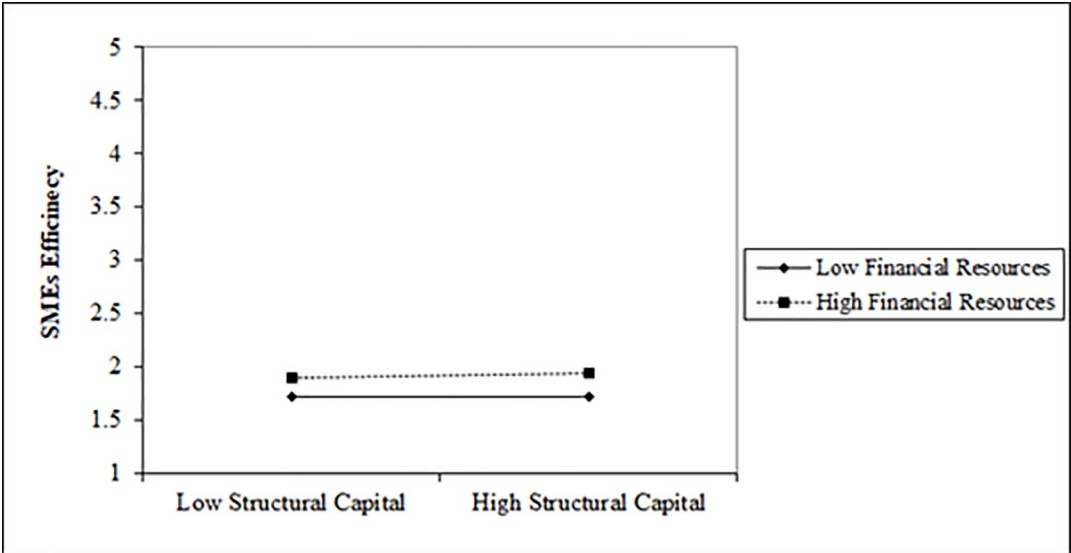

**Fig 5.**

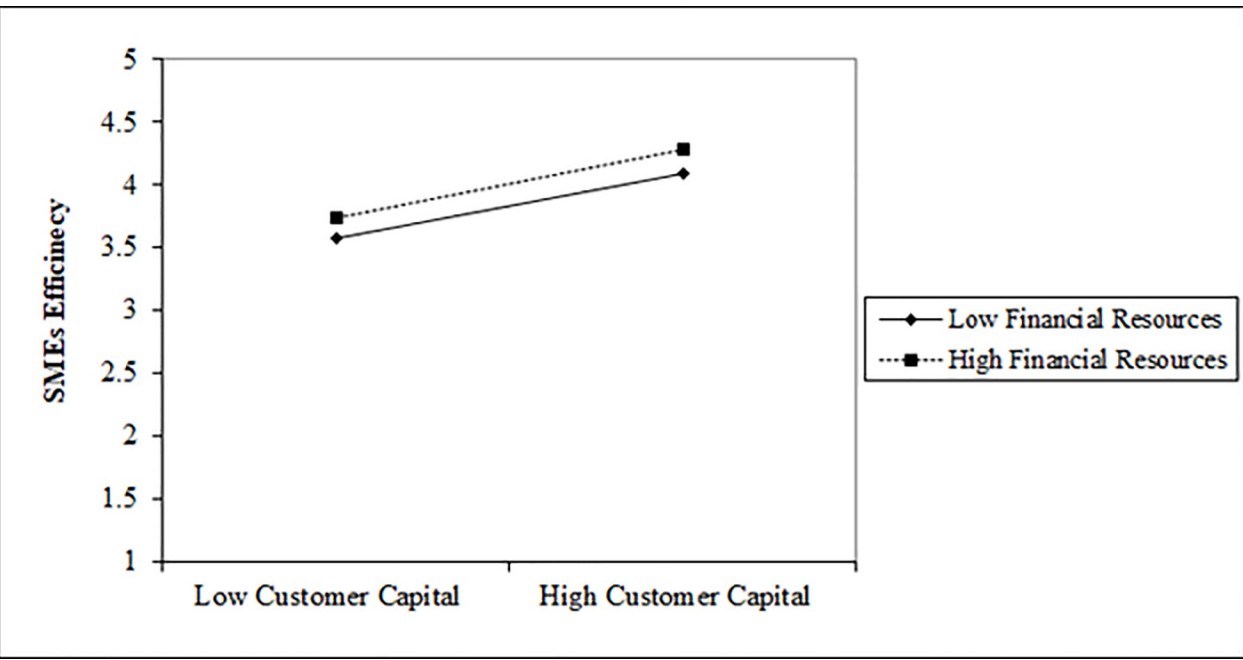

**Fig 6.**

5. Generally, do you agree that your performance and efficiency depend on IC and financial resources?

6. Ans: Three managers were agreed and said that IC and financial resources are the key to performance and efficiency in the current competitive markets. However, two managers said that other types of resources; IT, online business activities and reputation.

7. To summarize the interview results, we hereby confirm that overall, the results match with the survey data where the dimensions of IC and financial resources are considered crucial factors of high efficiency and performance.

## Discussion and conclusion

Based on the importance of intangible resources over tangible, this research tested the influence of each dimension of IC on firm efficiency with a moderating role of financial resources. Our findings have new implications for practicing managers and policymakers. These implications are different from previous studies. For instance, in the past, one zone of the research relied on tangible resources [108–110], while others, especially more recently give more weightage to intangible resources [2, 3, 45] in term of high performance and growth. This research collected empirical evidence from an emerging economy to discover how the intangible resources; human capital, customer capital and relational capital influence firm efficiency and how financial resources affect the relationship. Consequently, this research favors RBV theory where more weightage is given to intangible resources over tangible resources toward competitive advantage and superior performance. Though, the prior studies have mostly focused on the relationship between intangible resources and competitive advantage and performance. However, this research assessed the relationship between each dimension of IC and firm efficiency. In this research, we argue that IC—being an intangible resource significantly

**Table 6. Regression analysis.**

| | | Coefficients | | | | | | |
|---|---|---|---|---|---|---|---|---|
| Model | | Unstandardized Coefficients | | Standardized Coefficients | t | Sig. | | |
| | | B | Std. Error | Beta | | | $R^2$ | $R^2\Delta$ |
| 1 | (Constant) | 2.396 | 0.119 | | 20.132 | 0.000 | 0.058 | 0.058 |
| | Size | 0.013 | 0.021 | 0.038 | 0.631 | 0.529 | | |
| | Age | 0.100 | 0.036 | 0.169 | 2.771 | 0.006 | | |
| | education | 0.074 | 0.031 | 0.146 | 2.405 | 0.017 | | |
| 2 | (Constant) | 1.231 | 0.324 | | 3.804 | 0.000 | 0.120 | 0.063 |
| | Size | 0.023 | 0.021 | 0.067 | 1.115 | 0.266 | | |
| | Age | 0.096 | 0.036 | 0.161 | 2.673 | 0.008 | | |
| | education | 0.064 | 0.030 | 0.125 | 2.099 | 0.037 | | |
| | HC | 0.171 | 0.065 | 0.167 | 2.609 | 0.010 | | |
| | SC | -0.029 | 0.062 | -0.030 | -0.468 | 0.641 | | |
| | CC | 0.114 | 0.076 | 0.105 | 1.504 | 0.134 | | |
| | FR | 0.073 | 0.052 | 0.087 | 1.408 | 0.160 | | |
| 3 | (Constant) | 1.294 | 0.339 | | 3.818 | 0.000 | 0.152 | 0.032 |
| | Size | 0.026 | 0.021 | 0.073 | 1.244 | 0.215 | | |
| | Age | 0.090 | 0.036 | 0.151 | 2.525 | 0.012 | | |
| | education | 0.075 | 0.030 | 0.148 | 2.501 | 0.013 | | |
| | HC | 0.183 | 0.065 | 0.179 | 2.794 | 0.006 | | |
| | SC | -0.110 | 0.070 | -0.114 | -1.580 | 0.115 | | |
| | CC | 0.072 | 0.086 | 0.066 | 0.837 | 0.403 | | |
| | FR | 0.041 | 0.054 | 0.049 | 0.760 | 0.448 | | |
| | HCxFR | -0.032 | 0.019 | -0.182 | -1.714 | 0.088 | | |
| | SCxFR | 0.045 | 0.022 | 0.235 | 2.048 | 0.042 | | |
| | CCxFR | 0.022 | 0.021 | 0.124 | 1.063 | 0.289 | | |

a. Dependent Variable: Efficiency

contributes to efficiency when ventures have sufficient financial resources. As pointed out by the RBV theory, both tangible and intangible resources are crucial for venture success and profitability. Our study scrutinized that only intangible resources are not worthy, but there should be sufficient tangible resources (e.g. finance) to enjoy high efficiency in the competitive market.

We found that human capital significantly enhances firm efficiency that supported H1 of the research. In line with Mehralian et al., [111] who scrutinized that human capital is the most significant predictor of competitive advantage in the business industry. Similarly, Veltri and Silvestri [112] argued that human capital is a noteworthy factor in high growth and value in business firms. However, our study shows that financial resources significantly reduces the influence of human capital on SMEs' efficiency that is different from other studies. Our study opposes Chen, Hsu and Chang [113] who claimed that human capital helps managers to manage resources that in turn spurs their degree of internationalization and firm growth. Similarly, our findings are different from Delery and Roumpi [114] who claimed that human capital provides a sustainable competitive advantage in the presence of sufficient resources and valuable opportunities. The reason for the negative moderation is that managers in SMEs are unable to use financial resources efficiently due to a lack of financial skills. For instance, Klapper, Lusardi and Van Oudheusden [115] claimed that the lowest rate of financial literacy is reported in emerging economies such as Pakistan and China. Another reason that the majority of SMEs

are controlled by agents rather managed by the owners directly. Perhaps the agents do not effectively manage the financial resources for operational activities.

Our study displayed that structural capital directly does not improve SMEs' efficiency unless the ventures have sufficient FR. In contrast to Long Kweh, Lu and Wang [70] who scrutinized that structural capital significantly contributes to SMEs' efficiency. Our study also opposes Andreeva and Garanina [116] who exhibited a significant positive association between structural capital and values in Russian firms. However, our findings are related to Bontis, Ciambotti, Palazzi and Sgro [117] who exposed that structural capital does not significantly influence corporate social performance. However, we found that financial resources significantly reinforce the impact of structural capital on SMEs' efficiency in the emerging market China. Our findings support Buckley et al., [87] who established that financial resources enable firms to manage and structure internal resources articulately that result in high efficiency and growth.

Our analysis revealed that customer capital does not directly mend SMEs' efficiency in emerging markets. Unlike Yang and Kang [118] who reported a significant positive association between customer capital and business profitability, our study displays an insignificant path in emerging SMEs. However, our findings favor Bianchi Martini, Corvino, Doni and Rigolini, [119] who did not find any significant association between relational capital and enterprises value. Similarly, Andreeva and Garanina [116] also did not scrutinize any significant connection between relational capital and business profitability. Our findings partially favor Pedro, Leitão and Alves [120] who scrutinized that customer capital positively influences profitability but this relationship is fortified in the existence of other factors, resources and surroundings environment. In a similar vein, Luo, Griffith, Liu and Shi [121] exposed that HC, structural capital and customer capital influence financial performance but these links are moderated by ownership of the business. We argue that the direct relationship between IC and SMEs' efficiency is questionable, it can be moderated by venture internal resources.

## Implications for practice

This research provides imperative operational guidelines for managers and owners of SMEs to improve their venture efficiency through IC and FR. First, SMEs should improve their human capital as it directly configures business efficiency in the dynamic markets. However, this study revealed that financial resources weaken the influence of human capital on SMEs' efficiency, hence it the management of financial resources should be investigated. Perhaps managers cannot effectively use the financial resources for the operational activities due to a lack of financial skills. Therefore, it is owners to assign financial investment to financial literature managers, because they can manage the capital in a better way [122]. Another suggestion derived from this research is that managers with interpersonal human skills and with a lack of financial skills should not take responsibility for capital investment. They should use non-financial resources and information to enhance their venture efficiency. However, our study displays that structural capital and customer capital do not directly improve SMEs' efficiency unless there are adequate financial resources existed. Hence, ventures need to bring sufficient finance to boost their operational activities because mere structural capital and customer capital do not help in high efficiency. SMEs should not rely directly on their structural capital and customer capital but they should first manage sufficient finance to get maximum benefits and efficiency. SMEs should not exclusively use structural capital and customer capital for efficiency goal but managers/owners should balance their finance that can strengthen the path between structural capital, customer capital and efficiency. It is also suggested for SMEs that financial capital should not be directly invested in projects because it does not significantly

enhance efficiency. Ventures need to use the financial resources as well as structural capital and customer capital complementary to configure their efficiency. The implications are not limited to SMEs but listed firms can also apply it for their objectives and goals. Similarly, venture trade in developed markets can also get advantages of this study by modifying their strategic posture in terms of IC and capital.

Some of the key implications are highlighted below.

➢ SMEs need to use their human capital exclusively for enjoying high efficiency in the turbulent market. They should not use financial resources and human capital combine to getting efficiency and profitability in the emerging market China.

➢ However, structural capital and customer capital should not be used exclusively, but ventures should manage sufficient finance to get satisfactory efficiency via structural capital and customer capital.

➢ Financial resources do not enhance efficiency directly, but it must be used with structural capital and structural capital to acquire a desirable advantage.

## Limitations and future research

Despite having significant contributions of this research to RBV theory and the literature of IC, financial resources and firm efficiency, this research has some limitations that are needed to be addressed in future studies. The model is tested in the transition market China that may not be deemed the best representative of other emerging markets—hereby suggested to extend this model in other emerging markets. Moreover, developed markets can be confirmed to enhance the validity of the model if intangible resources are very prominent for SMEs' efficiency. This research is based on empirical evidence that are collected from SMEs through a structured questionnaire, future researchers can use open-ended questions and interviews (with several managers instead of only a few, as done in this study) that can give useful information in this nature of studies. Moreover, to reduce social desirability bias, the survey can be filled from different source or other person and other control factors can be applied to reduce the chances of spurious results. Large firms can be surveyed or data from financial reports and statements can be collected for testing the model. In this study, the only moderating role of financial resources is checked. Though, the other resources and capabilities such as technology and market knowledge can give fruitful insights. Moreover, to avoid common method bias, a longitudinal study can be conducted in future studies. We tested the moderating role of financial resources between the dimensions of IC and SMEs efficiency and revealed unexpected results. We suggest considering the financial knowledge and financial literacy of owners/managers to discern if it is worthy of high efficiency in the presence of IC. We controlled the age and size of the enterprises in this research. However, it will better to unleash how small size, middle size and large ventures as well as new and matured ventures get benefits of IC in the presence of FR. We further suggest assessing the role in organization competitiveness, innovative and environmental performance if financial resources facilitate these outcomes.

## Conclusion

The present study tests the influence of the intangible resources (human capital, structural capital and customer capital) on SMEs' efficiency with a moderating role of financial resources. We surveyed 264 Chinese SMEs via a self-administrated questionnaire to test the model. We used AMOS to test the hypotheses and generate results. Our study exposed that only human capital directly improves SMEs' efficiency while structural capital and customer capital do not

directly spur efficiency. However, we found that financial resources as a moderator reduce the influence of human capital on efficiency while significantly improve the influence of structural and customer capital on SMEs' efficiency. Our study suggests that SMEs should use human capital exclusively while structural and customer capital should be merged with financial resources to get higher efficiency. Our study also advises the Chinese government to initiate financial programs for industrial growth, so the enterprises will able to borrow loan for their operational activities.

## Supporting information

**S1 Appendix.**
(DOCX)

## Author Contributions

**Conceptualization:** Guowei Li.

**Formal analysis:** Guowei Li, Zhe Luo.

**Funding acquisition:** Xiantao Wang.

**Investigation:** Xiantao Wang.

**Project administration:** Yuqiu Lu.

**Resources:** Yuqiu Lu.

**Validation:** Xuening Liu.

**Visualization:** Xuening Liu.

**Writing – original draft:** Muhammad Anwar.

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
