## [Decision Letter · Decision Letter 0]

20 Mar 2020

PONE-D-20-03377

Intellectual Capital and the Efficiency of SMEs in the Transition Economy China; Do Financial Resources Strengthen the Routes?

PLOS ONE

Dear Dr. Anwar,

Thank you for submitting your manuscript to PLOS ONE. After careful consideration, we feel that it has merit but does not fully meet PLOS ONE’s publication criteria as it currently stands. Therefore, we invite you to submit a revised version of the manuscript that addresses the points raised during the review process.

The reviewer 1 has minor concerns about your paper, but unfortunately reviewer 2 have some major concerns. However, after a careful reading of the comments I think that it will be easy to address the modifications. I suggest to focus on the reviewer concerns about methodology.

We would appreciate receiving your revised manuscript by May 04 2020 11:59PM. To enhance the reproducibility of your results, we recommend that if applicable you deposit your laboratory protocols in protocols.io, where a protocol can be assigned its own identifier (DOI) such that it can be cited independently in the future. For instructions see: http://journals.plos.org/plosone/s/submission-guidelines#loc-laboratory-protocols

We look forward to receiving your revised manuscript.

Kind regards,

J E. Trinidad Segovia

Academic Editor

PLOS ONE

Journal Requirements:

4. Thank you for including your ethics statement:  "N/A".   

5. Please ensure that you refer to Figure 3 in your text as, if accepted, production will need this reference to link the reader to the figure.

6. Thank you for stating the following financial disclosure: "NO"

a)    Please provide an amended Funding Statement that declares *all* the funding or sources of support received during this specific study (whether external or internal to your organization) as detailed online in our guide for authors at http://journals.plos.org/plosone/s/submit-now.  

b)    Please state what role the funders took in the study.  If any authors received a salary from any of your funders, please state which authors and which funder. If the funders had no role, please state: "The funders had no role in study design, data collection and analysis, decision to publish, or preparation of the manuscript."

7. In your Data Availability statement, you have not specified where the minimal data set underlying the results described in your manuscript can be found. PLOS defines a study's minimal data set as the underlying data used to reach the conclusions drawn in the manuscript and any additional data required to replicate the reported study findings in their entirety. All PLOS journals require that the minimal data set be made fully available. For more information about our data policy, please see http://journals.plos.org/plosone/s/data-availability.

Reviewers' comments:

Reviewer's Responses to Questions

**Comments to the Author**

1. Is the manuscript technically sound, and do the data support the conclusions?

Reviewer #1: Yes

Reviewer #2: Partly

2. Has the statistical analysis been performed appropriately and rigorously? 

Reviewer #1: Yes

Reviewer #2: Yes

3. Have the authors made all data underlying the findings in their manuscript fully available?

Reviewer #1: Yes

Reviewer #2: Yes

4. Is the manuscript presented in an intelligible fashion and written in standard English?

Reviewer #1: Yes

Reviewer #2: Yes

5. Review Comments to the Author

Reviewer #1: The manuscript holds twofold novel contributions, include;

1. Testing the relationship between intellectual capital and SMEs efficiency that is perhaps missed in the previous studies. As the relationship between intellectual capital and SMEs performance is widely discussed but the efficiency of SMEs is very rarely studied in existing researches. This is important for ventures competing in the transition economy China.

2. The relationship is tested in the presence of financial resources that gives a plus point to the novelty. Indeed both intangible (intellectual capital) and tangible (financial resources) are very important for a venture growth.

However, I have a few minor concerns in this research that are highlighted below;

• I see that major parts are already covered in this introduction but the authors need to discuss the policy implications in detail.

• Criteria for sampling approach is not well-explained, so it may be explained clearly.

• Procedure of the data collection need to be stated in an articulated way.

• Authors need to discuss why they control the particular variables.

• Please provide a short note under the table 4, so that the readers can understand the idea.

• In the discussion chapter, I suggest to provide a brief summary of major findings because it is very important in social science studies.

Additionally, several grammatical and English structural mistakes are found in this manuscript, that need to be corrected via professional proofreading.

The rest of the work is impressive, and the manuscript is recommended for possible acceptance after the minor correction are made accordingly.

Good luck.

Reviewer #2: Thanks for the opportunity to revise this paper.

This work explores the role of financial resources in firm efficiency as well as how financial resources strengthen the relation between IC and SMEs efficiency. I like the gap of the paper as well as the structure, development, and especially I think it contains good contributions, conclusions and implications. However, there are several changes that I have to suggest, in order to correct this version of the paper, because some of these contributions are questionable according with both theoretical and empirical questions.

Please, I invite the authors to reflect and modify the paper in the next questions.

INTRODUCTION:

The authors should emphaticise the importance of connecting IC dimensions’ to efficiency and competitive advantage and they should show more clearly, if they focus on efficiency or competitive advantage and why.

They should introduce more clearly, they chose resource base view and no other organizational theories for their research.

THEORETICAL BACKGROUND AND HYPOTHESES DEVELOPMENT

Please if you are going to use efficiency as a central key……..do not use it collaterally (eg. Pg 10). I advise you to introduce one or two paragraphs that explain the importance of efficiency in general and specifically in SMEs.

Please, I’d like to read how the authors use Resource based View with great depth, for example for explaining how IC includes or not the traditional requirements (VRIO) of this theory for achieve competitive advantage. It can help a lot to strengthen the H1, H2 and H3. In this version, the arguments are too general and without using the basic concepts of the theory.

Please, use more clearly the SMES’s characteristics for developing the hypothesis……….Certainly it could be one of the major contributions of the paper (SMEs context).

Regarding the moderated role of financial resources, the arguments are more clear and robust.

METHODOLOGY:

According the representativeness of the simple, how do you check the difference between the response and non-response firms?

Are you sure it is good idea to measure firm efficiency with self-reported answers of owners/managers?

How do you manage the social desirability bias? Why not do you use some other secondary variable to check unless efficiency?

You say: “GFI = 0.81, CFI = 0.87, TLI = 0.85 and NFI = 0.81 displayed acceptable values (closed to 0.90)” ……..Are you sure? I am not. These values are so lower for saying that they are acceptable values.

According validity and reliability of the variables, I see that in each variable there is some item that has a very low value (see table 4). Hc4 in Human Capital, sc2 in Structural Capital cc3 in Customer capital, fr4 in financial resources or fe7 in SMEs Efficiency.

How do you use control variables (size and age), as a categorical variable or continuous variable? If first, I think it is wrong to lose the information of having the continuous data.

Definitely, you have to improve the measurement instrument and to repeat analysis.

In regards to Figure 4,5,6……..which is p-value for the Gradient slope? It is not sufficient with showing the image.

In summary, this very good potential paper needs important improvements. The gap and the sample is appropriate, but both theoretical and empirical changes need to be done.

I encourage the authors to develop these changes in order to improve the paper. Good luck!

6. PLOS authors have the option to publish the peer review history of their article (what does this mean?). If published, this will include your full peer review and any attached files.

Reviewer #1: No

Reviewer #2: No

---

## [Author Response · Author response to Decision Letter 0]

20 Apr 2020

We are thankful to the reviewers and editor for the valuable comments on our manuscript. We have revised all the comments and have added worthy information to the research. We believe that the revised version is has much improved in terms of contents and validity. 

Review Comments to the Author

Reviewer #1: The manuscript holds twofold novel contributions, include;

1. Testing the relationship between intellectual capital and SMEs efficiency that is perhaps missed in the previous studies. As the relationship between intellectual capital and SMEs performance is widely discussed but the efficiency of SMEs is very rarely studied in existing researches. This is important for ventures competing in the transition economy China.

2. The relationship is tested in the presence of financial resources that gives a plus point to the novelty. Indeed both intangible (intellectual capital) and tangible (financial resources) are very important for a venture growth.

Ans: Thank you for the positive comment. 

However, I have a few minor concerns in this research that are highlighted below;

• I see that major parts are already covered in this introduction but the authors need to discuss the policy implications in detail.

Ans: Thank you for your comments. We have discussed policy implications in the introduction as shown in red color.

• Criteria for sampling approach is not well-explained, so it may be explained clearly.

Ans: Thank you. We have discussed it in detail. 

• Procedure of the data collection need to be stated in an articulated way.

Ans: We have discussed it in detail, please read red color

• Authors need to discuss why they control the particular variables.

Ans: We have added a new section where it is discussed why we use control variables.

• Please provide a short note under the table 4, so that the readers can understand the idea.

Ans: Thank you for your comments. We have written a short note below table 4.

• In the discussion chapter, I suggest to provide a brief summary of major findings because it is very important in social science studies.

Ans: Thank you, we have provided a short summary of major findings, please read red color text. 

Additionally, several grammatical and English structural mistakes are found in this manuscript, that need to be corrected via professional proofreading.

Ans: Thank you. We have proofread the manuscript from an English expert that can give clear description now. 

The rest of the work is impressive, and the manuscript is recommended for possible acceptance after the minor correction are made accordingly.

Good luck.

Reviewer #2: Thanks for the opportunity to revise this paper.

This work explores the role of financial resources in firm efficiency as well as how financial resources strengthen the relation between IC and SMEs efficiency. I like the gap of the paper as well as the structure, development, and especially I think it contains good contributions, conclusions and implications. However, there are several changes that I have to suggest, in order to correct this version of the paper, because some of these contributions are questionable according with both theoretical and empirical questions.

Please, I invite the authors to reflect and modify the paper in the next questions.

INTRODUCTION:

The authors should emphaticise the importance of connecting IC dimensions’ to efficiency and competitive advantage and they should show more clearly, if they focus on efficiency or competitive advantage and why.

Ans: Thank you. We have added a new paragraph in the introduction where the paths between the dimensions of IC and SMEs success have been stated. 

They should introduce more clearly, they chose resource base view and no other organizational theories for their research.

Ans: We have shed light on the RBV theory as it underlines our conceptual model because it convers tangible (financial resources) and intangible resources (IC). Moreover, several other studies have discussed the RBV theory when they tested the path between IC and SMEs performance.

Khan, S. Z., Yang, Q., & Waheed, A. (2019). Investment in intangible resources and capabilities spurs sustainable competitive advantage and firm performance. Corporate Social Responsibility and Environmental Management, 26(2), 285-295.

Anwar, M., Khan, S. Z., & Khan, N. U. (2018). Intellectual capital, entrepreneurial strategy and new ventures performance: Mediating role of competitive advantage. Business and Economic Review, 10(1), 63-93.

THEORETICAL BACKGROUND AND HYPOTHESES DEVELOPMENT

Please if you are going to use efficiency as a central key……..do not use it collaterally (eg. Pg 10). I advise you to introduce one or two paragraphs that explain the importance of efficiency in general and specifically in SMEs.

Ans: Thank you for the useful suggestion. We have added a new paragraph stating three types of efficiency and how these are importance in SMEs sector, please read under the literature section. 

Please, I’d like to read how the authors use Resource based View with great depth, for example for explaining how IC includes or not the traditional requirements (VRIO) of this theory for achieve competitive advantage. It can help a lot to strengthen the H1, H2 and H3. In this version, the arguments are too general and without using the basic concepts of the theory.

Please, use more clearly the SMES’s characteristics for developing the hypothesis……….Certainly it could be one of the major contributions of the paper (SMEs context).

Ans: Thank you for your comments. We have discussed the theory in detail and how it is matched our model and why SMEs need IC are discussed in the literature section, please read red text. 

Regarding the moderated role of financial resources, the arguments are more clear and robust.

Ans: Thank you. 

METHODOLOGY:

According the representativeness of the simple, how do you check the difference between the response and non-response firms?

Ans: Thank you for pointing out this issue. We have executed a new test to check the problem that have discussed. We are aware which firm provided early response and which is late, because as we received our information, we are continuously entered in the SPSS file. 

Are you sure it is good idea to measure firm efficiency with self-reported answers of owners/managers?

Ans: We appreciate the point raised by the reviewer. Indeed we have faced this question many time in top ranked journals. We have discussed in the manuscript why self-reported measures are better in case of SMEs. Please read the measurement section and particularly “efficiency”

How do you manage the social desirability bias? Why not do you use some other secondary variable to check unless efficiency?

Ans: Thank you for the comment. We have mentioned why efficiency is self-reported measured and have provided evidence based reasons under the measurement section. Please read the red part.

You say: “GFI = 0.81, CFI = 0.87, TLI = 0.85 and NFI = 0.81 displayed acceptable values (closed to 0.90)” ……..Are you sure? I am not. These values are so lower for saying that they are acceptable values.

Ans: The reviewer has raised a valid point. There is no universal criteria for a model fit. However, values greater than 0.80 are considered acceptable when CMIN/DF is less than 3. 

The values greater than 0.80 are mentioned in the following studies;

• Wales, W. J., Patel, P. C., & Lumpkin, G. T. (2013). In pursuit of greatness: CEO narcissism, entrepreneurial orientation, and firm performance variance. Journal of Management Studies, 50(6), 1041-1069.

• Panigyrakis, G. G., & Theodoridis, P. K. (2007). Market orientation and performance: An empirical investigation in the retail industry in Greece. Journal of Retailing and Consumer Services, 14(2), 137-149.

• Liu, C. H. (2017). The relationships among intellectual capital, social capital, and performance-The moderating role of business ties and environmental uncertainty. Tourism Management, 61, 553-561.

According validity and reliability of the variables, I see that in each variable there is some item that has a very low value (see table 4). Hc4 in Human Capital, sc2 in Structural Capital cc3 in Customer capital, fr4 in financial resources or fe7 in SMEs Efficiency.

Ans: Again a valid question. However, I would like to discuss it for clarity. Hair et al., (2006-2010) stated that an item with standard factor loading above 0.70 should be retained and below 0.40 should be dropped. However, items in range of 0.41-0.69 should be retained or dropped after calculating their average. For instance, if there are three items with 0.80, 0.50 and 0.90 loading respectively. The average value of these three factor is 0.70, hence the item with a lower factor loading (0.50) can be retained. On the basis of this suggestion, we have retained the items. The same approach is followed in several studies such as

Anwar, M., Rehman, A. U., & Shah, S. Z. A. (2018). Networking and new venture’s performance: Mediating role of competitive advantage. International Journal of Emerging Markets.

Khan, S. Z., Yang, Q., & Waheed, A. (2019). Investment in intangible resources and capabilities spurs sustainable competitive advantage and firm performance. Corporate Social Responsibility and Environmental Management, 26(2), 285-295.

How do you use control variables (size and age), as a categorical variable or continuous variable? If first, I think it is wrong to lose the information of having the continuous data.

Ans: Thank for the worthy comment. We have discussed the control variables in detail. Please read the methodology section where it is stated that nature of industry is a categorical variable and size and age of firms as well as educational background are continuous variables that are controlled directly in structural model. 

In regards to Figure 4,5,6……..which is p-value for the Gradient slope? It is not sufficient with showing the image.

Ans: Thank you. We have discussed the p values, highlighted red. 

In summary, this very good potential paper needs important improvements. The gap and the sample is appropriate, but both theoretical and empirical changes need to be done.

Ans: Thank you for the comment. We have discussed theoretical section in detail and have applied regression analysis in SPSS to check if there is difference in the results. 

I encourage the authors to develop these changes in order to improve the paper. Good luck!

Thank you once again for your time and consideration.

---

## [Decision Letter · Decision Letter 1]

11 May 2020

PONE-D-20-03377R1

Intellectual Capital and the Efficiency of SMEs in the Transition Economy China; Do Financial Resources Strengthen the Routes?

PLOS ONE

Dear Dr. Anwar,

Thank you for submitting your manuscript to PLOS ONE. After careful consideration, I feel that it has merit but does not fully meet PLOS ONE’s publication criteria as it currently stands. Therefore, I invite you to submit a revised version of the manuscript that addresses the points raised during the review process.

I consider that the manuscript is almost ready for publication because most of the comments have been attended.  However, one of the reviewers still have some concerns that need to be addressed in a new revised version.

I would appreciate receiving your revised manuscript by Jun 25 2020 11:59PM. To enhance the reproducibility of your results, we recommend that if applicable you deposit your laboratory protocols in protocols.io, where a protocol can be assigned its own identifier (DOI) such that it can be cited independently in the future. For instructions see: http://journals.plos.org/plosone/s/submission-guidelines#loc-laboratory-protocols

We look forward to receiving your revised manuscript.

Kind regards,

J E. Trinidad Segovia

Academic Editor

PLOS ONE

Reviewers' comments:

Reviewer's Responses to Questions

**Comments to the Author**

1. If the authors have adequately addressed your comments raised in a previous round of review and you feel that this manuscript is now acceptable for publication, you may indicate that here to bypass the “Comments to the Author” section, enter your conflict of interest statement in the “Confidential to Editor” section, and submit your "Accept" recommendation.

Reviewer #1: (No Response)

Reviewer #2: All comments have been addressed

2. Is the manuscript technically sound, and do the data support the conclusions?

Reviewer #1: Yes

Reviewer #2: Yes

3. Has the statistical analysis been performed appropriately and rigorously? 

Reviewer #1: Yes

Reviewer #2: N/A

4. Have the authors made all data underlying the findings in their manuscript fully available?

Reviewer #1: Yes

Reviewer #2: Yes

5. Is the manuscript presented in an intelligible fashion and written in standard English?

Reviewer #1: Yes

Reviewer #2: Yes

6. Review Comments to the Author

Reviewer #1: The manuscript may thoroughly be checked before online publication by the journal in order to remove any minor language or technical mistakes.

Reviewer #2: Thank you very much for the efforts in driving all the recommendations made in my first review.

I feel that most of the recommendations have been well conducted. However, there are 2 issues that I consider have not yet been resolved by the authors. These two questions are relevant, especially the second, and their resolution would increase the robustness of the results of the work.

These questions are the following:

First, I still disagree on the use of size and age variables as categorical and not as continuous variables. I believe that this issue remains unresolved and I suppose I think this question can be solved, probably with a simple phone call to the company, for example. The use of these variables as categorical reduce their explanatory power of the model even more in a sample that is already SMES.

Second, I recognize that there are papers where efficiency can be measured with self-reported methods. But it is also evident that the limitation of work is then very important especially if other control measures are not taken and there is clearly an effect of social desirability bias. There are other steps that can be taken to address this issue, such as surveying performance issues from a different source (for example, other person).

I encourage authors to make one last effort to resolve these issues, especially the issue of social desirability bias.

Good luck!

7. PLOS authors have the option to publish the peer review history of their article (what does this mean?). If published, this will include your full peer review and any attached files.

Reviewer #1: No

Reviewer #2: No

---

## [Author Response · Author response to Decision Letter 1]

14 May 2020

We are thankful to the reviewers and editor for the valuable comments on our manuscript. We have revised all the comments and have added worthy information to the research. We believe that the revised version contains worthy implications and is ready for publication. 

 Reviewer #2: All comments have been addressed

All the comments are addressed. 

Ans: Thank you very much for the time and suggestions of the reviewer that enabled us in building a quality research work.

Reviewer #1:

I feel that most of the recommendations have been well conducted. However, there are 2 issues that I consider have not yet been resolved by the authors. These two questions are relevant, especially the second, and their resolution would increase the robustness of the results of the work.

These questions are the following:

First, I still disagree on the use of size and age variables as categorical and not as continuous variables. I believe that this issue remains unresolved and I suppose I think this question can be solved, probably with a simple phone call to the company, for example. The use of these variables as categorical reduce their explanatory power of the model even more in a sample that is already SMES.

Ans: We appreciate the point raised by the reviewer to reduce spurious results and increase explanatory power of the model. 

However, it is a challenge for us to know when the venture is started and how many people are exactly working in the firms. Because the survey is conducted using a hard copy approach before the COVID-19 crises and it was easy for us to go and connect with responsible managers. Since the crises have changed the situation and enterprises have not yet formally started their operational activities physically but they are working virtually. We have tried to contact the enterprises who have participated in the survey but contacting via call did not favor. We have called several firms, in most of the cases, the bottom level managers are connected and excused to talk top management team. Unfortunately, response of the bottom level managers was not satisfactory for us because only top managers and owners understand strategic policies, performance activities and decision making process of their firms (Degong et al., 2018; Songling et al., 2018; Tajeddini and Milluer, 2012). 

We realized that it is big challenge and perhaps impossible to reach and connect with all the enterprises (264) in Beijing, Shanghai and Shenzhen (e.g. due to the current situation-COVID-19) who have already provided their responses via a hard copy questionnaire. 

Additionally, we have read a few studies in similar nature who have also used the same strategy for age and size of the enterprises. 

1. Khan, S. Z., Yang, Q., Khan, N. U., Kherbachi, S., & Huemann, M. (2020). Sustainable social responsibility toward multiple stakeholders as a trump card for small and medium‐sized enterprise performance (evidence from China). Corporate Social Responsibility and Environmental Management. 

2. Memon, A., Yong An, Z., & Memon, M. Q. (2020). Does financial availability sustain financial, innovative, and environmental performance? Relation via opportunity recognition. Corporate Social Responsibility and Environmental Management, 27(2), 562-575.

3. Khan, S. Z., Yang, Q., & Waheed, A. (2019). Investment in intangible resources and capabilities spurs sustainable competitive advantage and firm performance. Corporate Social Responsibility and Environmental Management, 26(2), 285-295.

4. Ilyas, S., Hu, Z., & Wiwattanakornwong, K. (2020). Unleashing the role of top management and government support in green supply chain management and sustainable development goals. Environmental Science and Pollution Research, 1-14.

5. Anwar, M. (2018). Business model innovation and SMEs performance—Does competitive advantage mediate?. International Journal of Innovation Management, 22(07), 1850057.

Second, I recognize that there are papers where efficiency can be measured with self-reported methods. But it is also evident that the limitation of work is then very important especially if other control measures are not taken and there is clearly an effect of social desirability bias. There are other steps that can be taken to address this issue, such as surveying performance issues from a different source (for example, other person).

Ans: Thank you once again for raising the valid comment of social desirability bias in performance measures. The reviewer is right that there are papers that have measured the efficiency as a self-reported method. This is especially true in case of SMEs because they do not provide their financial information to the general public, as stated in the manuscript. Also this claim is stated in several research papers measuring efficiency:

1. Khattak. M.S. Shah. S.Z.Z (2020). Top Management Capabilities and Firm Efficiency: Relationship via Resources Acquisition. Business and Economic Review, 12(1), 87-118. https://imsciences.edu.pk/files/journals/vol12_2020/New%205-%20667.pdf

2. Martin, S. L., & Javalgi, R. R. G. (2016). Entrepreneurial orientation, marketing capabilities and performance: the moderating role of competitive intensity on Latin American International new ventures. Journal of Business Research, 69(6), 2040-2051.

To avoid the problem, we have already applied a few measures; validity, reliability and common method variance that are recommended in other studies, listed above. Additionally, we requested owners and top managers as they are the people who know the financial position and strategic activities of their firms. Additionally, where managers were not aware of exact financial figures, they recommended us to ask financial managers regarding efficiency and financial outcomes. Not in all the firms but in most cases, the survey is not filled by a single person, especially in those enterprises where agents were working of the behalf of owners of the firms. 

We asked them that the survey is volunteer to be filled. To reduce social desirability bias, we have clearly mentioned in the covered letter of the survey that the data of this survey is exclusively used for research analysis, and the information will not be shared elsewhere.

Additionally, we read a few papers where they have used open ended questions/interview as a robustness check for mitigating social desirability bias and improving validity of the results. 

• Anwar, M., Khan, S. Z., & Khan, N. U. (2018). Intellectual capital, entrepreneurial strategy and new ventures performance: Mediating role of competitive advantage. Business and Economic Review, 10(1), 63-93.

• Saad, A., Xinping, G., & Ijaz, M. (2019). China-Pakistan Economic Corridor and its influence on perceived economic and social goals: Implications for social policy makers. Sustainability, 11(18), 4949.

Perhaps this was a suitable method for us to follow this approach. Parsimoniously, we relied on this approach and conducted a brief interview with five firms that has discussed below.

Online interviews

We conducted an in-depth interview with five managers/owners of the SMEs to enhance validity of the results and especially to know if there is social desirability bias in the data. We asked the following questions in the interview to get open ended answers.

1. Elaborate your IC strength in terms of human capital, structural capital and customers’ capital and how it benefits your firms, especially performance and efficiency?

Ans: Three managers shed light on the importance of their IC for high performance and efficiency while two firms replied that their IC averagely contribute to their performance. 

2. How your existing financial resources moderate the path between IC and efficiency?

Ans: Four managers said that they get high benefits of IC when financial resources work as a moderator while one manager said that my firm faces financial constraint, and we are unable to get desirable efficiency. 

3. Generally, do you agree that your performance and efficiency depend on IC and financial resources?

Ans: Three managers were agreed and said that IC and financial resources are the key to performance and efficiency in the current competitive markets. However, two managers said that other types of resources; IT, online business activities and reputation. 

To summarize the interview results, we hereby confirm that overall, the results match with the survey data where the dimensions of IC and financial resources are considered crucial factors of high efficiency and performance. 

I encourage authors to make one last effort to resolve these issues, especially the issue of social desirability bias.

Once again, we appreciate the points raised by the reviewer and advancing our knowledge to mitigate this issue in future studies.

Additionally, the reviewer is also welcome if she/he still sees elusiveness in the results. As realized in the online survey during the COVID-19, we will have only option to collect new data from random enterprises that may or not may include the firms who have participated in the structured survey via a hard copy approach. In this case, we have to analyze the results again because of new data, and see what are the new results are, perhaps similar or different. So we can take time to conduct the online survey due to COVID-19 in order to complete the analysis in a new way. 

We hope that the reviewer will agree and favor our comment. Thank you once again for your time and comments that enabled us in building a wonderful piece of research.

---

## [Decision Letter · Decision Letter 2]

17 Jun 2020

Intellectual Capital and the Efficiency of SMEs in the Transition Economy China; Do Financial Resources Strengthen the Routes?

PONE-D-20-03377R2

Dear Dr. Anwar,

We’re pleased to inform you that your manuscript has been judged scientifically suitable for publication and will be formally accepted for publication once it meets all outstanding technical requirements.

Kind regards,

J E. Trinidad Segovia

Academic Editor

PLOS ONE

Additional Editor Comments (optional):

Reviewers' comments:

Reviewer's Responses to Questions

**Comments to the Author**

1. If the authors have adequately addressed your comments raised in a previous round of review and you feel that this manuscript is now acceptable for publication, you may indicate that here to bypass the “Comments to the Author” section, enter your conflict of interest statement in the “Confidential to Editor” section, and submit your "Accept" recommendation.

Reviewer #2: (No Response)

Reviewer #3: All comments have been addressed

2. Is the manuscript technically sound, and do the data support the conclusions?

Reviewer #2: Partly

Reviewer #3: Partly

3. Has the statistical analysis been performed appropriately and rigorously? 

Reviewer #2: No

Reviewer #3: Yes

4. Have the authors made all data underlying the findings in their manuscript fully available?

Reviewer #2: No

Reviewer #3: Yes

5. Is the manuscript presented in an intelligible fashion and written in standard English?

Reviewer #2: Yes

Reviewer #3: Yes

6. Review Comments to the Author

Reviewer #2: Dear authors,

Thanks for your responses to my second revision.

I have to say that no one of the requested changes have been developed.

I'm not sure if it is rigorous justifying not to have secondary data of the sample because of the Covic. There are others sources (see for example the JBR's paper cited in the revision that does a lot of checking process in order to validate the response of the survey with secondary data).

According social desirability bias, authors haven't done anything either, only to justify it with 2 papers (one of them in a journal with much less evidence of quality than Plos One and other with one paper in JBR incorrectly cited, because JBR's paper has a lot of efforts to check that social desirability bias doesn't happen).

Unfortunately I have to say that the problems of the paper have not been resolved.

Sorry but I was expected to be something more in the paper. The balance between theoretical and empirical part is not sufficient.

Regards.

Reviewer #3: (No Response)

7. PLOS authors have the option to publish the peer review history of their article (what does this mean?). If published, this will include your full peer review and any attached files.

Reviewer #2: No

Reviewer #3: No

---

## [Editor Report · Acceptance letter]

19 Jun 2020

PONE-D-20-03377R2 

Intellectual Capital and the Efficiency of SMEs in the Transition Economy China; Do Financial Resources Strengthen the Routes? 

Dear Dr. Anwar:

I'm pleased to inform you that your manuscript has been deemed suitable for publication in PLOS ONE. Congratulations! Your manuscript is now with our production department. 

Kind regards, 

on behalf of

Dr. J E. Trinidad Segovia 

Academic Editor

PLOS ONE